# Depression Moderates the Relationship between Trait Anxiety, Worry and Attentional Control in Melanoma Survivors

**DOI:** 10.3390/healthcare11233097

**Published:** 2023-12-04

**Authors:** Elizabeth J. Edwards, Khanh Linh Chu, Nikeith John, Mark S. Edwards, Michael Lyvers

**Affiliations:** 1School of Education, The University of Queensland, St. Lucia, QLD 4072, Australia; linh.chu@uq.edu.au; 2Faculty of Health Sciences and Medicine, Bond University, Robina, QLD 4226, Australia; 3School of Education, Language and Psychology, York St. John University, York YO31 7EX, UK; m.edwards@yorksj.ac.uk; 4School of Psychology, Bond University, Robina, QLD 4226, Australia; mlyvers@bond.edu.au

**Keywords:** anxiety, attention, attentional control, attentional control theory, cancer-related worry, depression, melanoma survivors

## Abstract

Cancer survivors commonly contend with concurrent cognitive difficulties such as problems with attention and concentration, and psychological distress, including anxiety and depression. However, the associations between attentional and emotional difficulties within the specific context of melanoma survivors remain relatively unexplored. Premised on attentional control theory, the current study employed a cross-sectional design to explore the interplay among trait anxiety (dispositional) and situational anxiety (cancer-related worry), depression and attentional control (ability to inhibit distractors and flexibly shift within and between tasks) in a sample of 187 melanoma survivors aged 18 to 58 years (*M*_age_ = 36.83 years, *SD*_age_ = 5.44 years; 93% female). Data were analyzed using a moderated multiple regression, with anxiety, cancer worry and depression as predictors, and attentional control as the criterion variable. After statistically controlling for the variance of chemotherapy, we found that individuals with higher trait anxiety and higher cancer-related worry reported greater attentional control at low levels of depression, yet poorer attentional control at high depression, relative to individuals with low anxiety. Our findings suggest that anxiety and depression are differentially related to attentional control in melanoma survivors. The results provide a marker for clinicians addressing anxiety and depression in this population. Implications for primary healthcare are discussed.

## 1. Introduction

Advancements in melanoma treatment options have lowered mortality (five-year survival rate of 98%) and resulted in a large proportion of survivors living with the long-term sequelae of emotional distress [1,2,3,4]. One study with long-term cancer survivors, including those with melanoma, underscored the prevalence of comorbid anxiety and depression, with 87% of those reporting high anxiety also exhibiting increased depressive symptoms and 46% of those reporting high depression also reporting increased levels of anxiety [5]. Cancer-related worry is also common among melanoma survivors, evidenced by approximately 74% expressing fear of recurrence [6,7]. Furthermore, female melanoma survivors tended to report higher rates of anxiety, depression, and cancer-related worry compared to their male counterparts [8,9].

Cognitive problems such as attention and concentration difficulties are also reported in the broader cancer survivorship literature [10,11], including investigations with melanoma patients [12]. Studies with healthy participants have shown that heightened anxiety is related to attention and memory deficits [13,14]. Anxiety places increased demand on the ability to focus and shift attention, namely attentional control. This prompts a plausible association between elevated levels of anxiety, possible co-morbid depression, worry regarding cancer recurrence and attention problems among melanoma survivors. However, to date, no research has examined these unique and combined relationships within the context of melanoma survivorship. The current study aims to address this gap.

Numerous studies have investigated attentional problems in cancer survivors. Von Ah et al. [15] noted attention difficulties via the use of interviews, while Lycke et al. [16] reported poor performance on a digit span task. Interviews and digit span tasks, however, lack the precision necessary to serve as pure indices of attentional control. Furthermore, both studies excluded participants with existing symptoms of anxiety and depression, thus making it difficult to examine the association between attentional and affective problems. Research on attentional problems in melanoma survivors is limited. Bartels et al. [17] found compromised attentional control operationalized by performance on a range of cognitive tasks; however, anxiety and depression were outside the scope of their study. Rogier et al. [18,19] found attentional problems in melanoma survivors using a battery of cognitive tasks [20] and assessed anxiety and depression using self-reported symptom scales. Rogier et al. noted that higher emotional symptoms were related to poorer attentional control, yet they did not examine the interplay between these factors.

Studies investigating the interrelationship between emotional symptoms and attentional control in cancer survivors are limited, and to date, no studies have explored this connection in the specific context of melanoma survivors. In one study, Krolak et al. [21] observed no relationship between anxiety (indexed by a symptom checklist) and cognitive performance (assessed using a self-reported measure) in lymphoma survivors. Another study found similar results in cervical cancer survivors [22]. However, it is noteworthy that the measures of cognitive functioning they used, the Functional Assessment of Cancer Therapy-Cognitive subscale (FACT-Cog; [21]) and the Cognitive Symptoms Checklist (CSC; [22]), may have lacked the sensitivity required to discern nuances in attentional control. Calvio et al. [23] observed heightened anxiety and depression (assessed using symptom checklists) was related to poorer attention (indexed by a self-reported cognitive symptom checklist) in breast cancer survivors compared to their non-cancer counterparts. However, they found no interactive relationship between anxiety, depression, and attention. Interestingly, no studies have included cancer-related worry, which is prevalent in cancer survivorship. We argue that cancer-related worry, a unique form of situational anxiety, deserves consideration when exploring the combined associations between anxiety–depression–worry and attentional control in melanoma survivors.

Attentional control theory [24] proposes that individuals with elevated trait (dispositional) and/or state (situational) anxiety direct their attention towards potentially threatening information, which reduces their ability to perform cognitive tasks. The theory suggests that in stressful situations, highly anxious individuals become aware of their cognitive deficits and recruit additional resources (e.g., motivational effort) to lessen performance deficits to the extent that they can sometimes outperform their less anxious counterparts. There is substantial evidence to support this notion in healthy adults [25,26,27]. For example, Spada et al. [27] explored the relationship between attentional control (indexed by the Attentional Control Scale, ACS [28]) and situational anxiety (measured using the State-Trait Inventory for Cognitive and Somatic Anxiety-State subscale, STICSA-SC [29]), and found that higher situational anxiety was related to poorer attentional control. Olafsson et al. [30] also used the ACS and reported compromised attentional focusing and shifting in individuals with higher trait anxiety and depression. Although these studies were undertaken in a non-clinical context, they provide the foundation for our work with melanoma survivors.

While research suggests that anxiety interferes with the ability to focus/direct attention, depression appears to hinder the ability to inhibit worrisome thoughts and shift attention to task demands [30,31]. We propose that, in the case of melanoma survivors grappling with elevated anxiety (potentially both trait anxiety and situation-specific cancer worry), they may employ additional motivational effort to compensate for attentional deficits, potentially outperforming those who are less anxious. However, their depression may impede their ability to disengage from worrisome thoughts, potentially affecting their performance compared to their less depressed counterparts.

The present study examined the link between trait anxiety (measured using the Trait Cognitive subscale of the STICSA [29]), situational anxiety, i.e., cancer worry (measured using an adapted version of the Penn State Worry Questionnaire [32]), depression (assessed using the Depression subscale from the Depression Anxiety Stress Scale [33]), and attentional control (operationalized using the ACS [28]), in a sample of melanoma survivors. Given that melanoma management often includes chemotherapy, known to impair cognitive function (i.e., ‘chemo-brain’ [34]), and that chemotherapy has also been linked to anxiety and depression [35], we deployed a statistical control for chemotherapy in our analyses, thus enabling the relationship between anxiety, depression, and attentional control to be examined after removing the variance related to chemotherapy. Understanding the discrete and dynamic relationships between anxiety, cancer worry, depression, and attentional control has the potential to provide clinical markers, such that clinicians treating melanoma survivors can prioritize the treatment of emotional and cognitive symptoms.

Our hypotheses were premised on attentional control theory [24]. After controlling for chemotherapy, we predicted a 3-way interaction between trait anxiety, cancer worry, and depression on attentional control, such that the effects of anxiety would be moderated by depression. At lower depression levels, we hypothesized that higher trait anxiety levels would be related to higher attentional control at higher cancer worry levels but not at lower cancer worry levels. At higher depression levels, we hypothesized that higher trait anxiety levels would be related to poorer attentional control at higher cancer worry levels but not at lower cancer worry levels.

## 2. Materials and Methods

### 2.1. Measures

#### 2.1.1. State-Trait Inventory for Cognitive and Somatic Anxiety

The Trait-Cognitive subscale of the State-Trait Inventory for Cognitive and Somatic Anxiety (STICSA-TC; [29]) was used to index symptoms of Trait (dispositional) and Cognitive (thoughts) anxiety (10 items). Participants responded to items about their feelings ‘in general’ on a 4-point Likert scale from 1 = almost never to 4 = almost always (e.g., *I feel agonized over my problems*; *I think that the worst will happen*). No items were reversed scored, and scores were summed with total scores ranging from 10 to 40. High scores were indicative of higher anxiety levels. The Trait-Cognitive subscale demonstrated excellent internal consistency, identified as α = 0.87 [29].

#### 2.1.2. Penn State Worry Questionnaire

The 16-item Penn state worry Questionnaire (PSWQ; [32]) was modified to assess the severity of cancer-related worry. Participants were instructed to respond to statements about their cancer using a 5-point Likert scale from 1 = not at all typical of me to 5 = very typical of me. The scale comprises 11 positively scored items, e.g., *Once I start worrying, I can’t stop*) and 5 negatively worded or reverse-scored items, e.g., *I do not tend to worry about things*). Total scores were summed and range from 16 to 80, with higher total scores reflecting greater levels of cancer-related worry. The PSWQ demonstrated excellent internal reliability α = 0.93 [32].

#### 2.1.3. Depression Anxiety Stress Scale

The 7-item Depression subscale from the Depression Anxiety Stress Scale (DASS-D; [36]) was used to index depression. Participants identified how much statements applied to them ‘in the past week’ using a 4-point Likert scale from 0 = did not apply to me at all to 3 = applied to me very much or most of the time (e.g., *I felt I had nothing to look forward to*; *I felt down-hearted and blue*). Scoring required items to be summed and multiplied by 2, resulting in scores from 0 to 42. Higher scores reflected higher depression levels. The depression subscale achieved excellent internal reliability α = 0.81 [37].

#### 2.1.4. Attentional Control Scale

The Attentional Control Scale (ACS; [33]) is a 20-item self-report instrument assessing an individual’s ability to focus and shift attention. Participants responded to 8 positively worded statements of everyday attention (4, 5, 9, 10, 13, 14, 17, 19, e.g., it is easy for me to read or write while I’m also talking on the phone) and 12 negatively worded statements (1, 2, 3, 6, 7, 8, 11, 12, 15, 16, 18, 20, e.g., *When I am working hard on something*, *I still get distracted*) on a 4-point scale of 1 = almost never to 4 = always. Total scores ranged from 20 to 80, with higher scores reflecting better attentional control. The ACS showed excellent internal reliability analyses α = 0.88 [28].

### 2.2. Procedure

Participants were recruited from the Qualtrics Database of pre-screened research volunteers. Participants were informed that ‘*It is anticipated that the data collected during this study will assist us in understanding the mechanisms that underpin affective and cognitive processes as this understanding can have key downstream consequences for behavior across the cancer recovery continuum.*’ Respondents used an online link to complete the questionnaires, which were presented in a fixed order: demographics, STICSA-TC, DASS-D, PSWQ, and ACS. The initial screen included an explanatory statement and an instruction advising volunteers that, by clicking to the next page, they were implying their consent to participate. The study was approved by the University’s Human Research Ethics Committee, and data were collected between 1 April 2016 and 30 September 2016. Completion of the questionnaires took approximately 10 min, and no monetary incentive was offered.

### 2.3. Participants

One hundred and eighty-seven melanoma survivors aged between 18 and 58 years (*M_age_* = 36.83, *SD_age_* = 5.44; 93% female) participated. The demographic characteristics of the sample are shown in Table 1.

### 2.4. Data Analysis

A cross-sectional design was used. A moderated multiple regression model was created comprising trait anxiety and cancer-related worry as predictor variables, depression as the moderator variable, and attentional control as the criterion or outcome variable. The interaction terms were formed using mean-centered scores. Analyses were conducted using SPSS version 27. Interactions in multiple linear regression with SPSS version 24 and Excel software (version 1.6) (IRSE; [36]) was used to interpret the interactions and conduct tests of simple slopes at high and low values on the predictor variables (calculated at ±1 SD from the mean score on each). The modelling was adjusted for chemotherapy. Assumptions of multiple linear regression were examined before finalizing the statistical model. For our linear regression with eight predictors, an a priori power analysis using G*Power 3.1 indicated that a sample of 160 would achieve a power of 0.95 at *p* < 0.05, considering a medium effect size (*f*^2^ = 0.15).

## 3. Results

### 3.1. Data Diagnostics and Assumption Checking

The data were screened for extreme outliers. Univariate outliers were considered significant with z-scores > 3.50, and multivariate outliers were identified with the use of Mahalanobis distance, such that cases with *p* < 0.001 were considered significant. Using these criteria, eight cases were excluded as univariate outliers (six outliers for worry, z-scores = 4.96, 4.49, 4.49, 3.89, 3.66, and 3.66, and two outliers for attentional control, z-scores = 4.68, and 4.06), and no cases were identified as multivariate outliers. The final data set met the assumptions of normality, linearity, homoscedasticity, and independence of observations, and tests for skewness and kurtosis were acceptable (*N* = 179).

### 3.2. Descriptive Statistics

The means, standard deviations, zero-order and inter-correlations among the predictors and criterion variables are shown in Table 2. As can be seen, trait anxiety was significantly positively related to worry and depression (i.e., higher trait anxiety was associated with greater worry and greater symptoms of depression), and worry was significantly positively related to depression (i.e., greater worry was associated with greater depression). Attentional control was significantly positively correlated with trait anxiety, worry and depression; however the strength of these associations was small.

### 3.3. Main Analyses

A moderated multiple regression was conducted to determine whether removing any variance related to chemotherapy, trait anxiety, worry, depression and their interactions predicted attentional control. The covariate (chemotherapy) was entered at Step 1, the component main effects (trait anxiety, worry and depression) were entered at Step 2, the two-way interaction terms (trait anxiety × worry, trait anxiety × depression, and worry × depression) were entered at Step 3, and the three-way interaction term (trait anxiety × worry × depression) was entered at Step 4.

The unstandardized coefficients, t values, *p* values, and 95% confidence intervals for all variables are shown in Table 3. At Step 1, chemotherapy accounted for 6% of the variance in attentional control, which was significant, *R* = 0.23, *F* (1, 177) = 10.25, *p* = 0.002. Chemotherapy was associated with better attentional control. The component main effects were added at Step 2, and the change *R*^2^ accounted for 8% of the variance in attentional control; however, *R* = 0.28, Δ*R*^2^ = 0.02, Δ*F* (3, 174) = 1.31, *p* = 0.272 were not significant, although the model, *F* (4, 174) = 3.56, *p* = 0.008, was significant. At Step 3, the addition of the two-way interaction terms did not produce a change in *R*^2^, *R* = 0.33, Δ*R*^2^ = 0.04, Δ*F* (3, 171) = 2.28, *p* = 0.081, and the model, accounting for 11% of variance in the criterion, remained significant, *F* (7, 171) = 3.06, *p* = 0.005. At Step 4, the three-way interaction term resulted in a significant increase in *R*^2^, *R* = 0.40, *F* (1, 170) = 10.49, *p* = 0.001, and the full model accounted for 16% of the variance in attentional control, which was significant, *F* (8, 170) = 4.13, *p* < 0.001.

Interactions in Multiple Linear Regression with SPSS and Excel software [IRSE; 37], was used to decompose the three-way interaction and perform tests of simple slopes at high and low values on the trait anxiety and depression scales (calculated at ±1 SD from the mean score on each). Figure 1 shows the pattern of the interaction. The left-hand panel shows that at low depression (−1 SD), attentional control varied as a function of trait anxiety and worry. At higher worry, higher trait anxiety was associated with better attentional control, β = 0.36, *t* = 4.71, *p* < 0.001; however, at lower worry, trait anxiety was unrelated to attentional control, β = 0.05, *t* < 1. The right-hand panel shows a markedly different relationship between trait anxiety and worry for high depression (+1 SD). There was again no relationship between trait anxiety and attentional control at lower worry β = 0.00, *t* < 1; however, at higher worry, higher trait anxiety predicted poorer attentional control, β = −0.33, *t* = 3.18, *p* = 0.002.

## 4. Discussion

The present study investigated the relationship between anxiety (trait anxiety and cancer-related worry), depression, and attentional control in melanoma survivors. Specifically, we predicted that, after controlling for chemotherapy, the effects of anxiety on attentional control would be moderated by depression. At lower depression, we hypothesized that higher trait anxiety would be related to higher attentional control at higher cancer worry levels but not at lower cancer worry levels. At higher depression, we hypothesized that higher trait anxiety would be related to poorer attentional control at higher cancer worry levels but not at lower cancer worry levels. Our results were consistent with these hypotheses.

The present study was the first to examine the link between emotional symptoms (anxiety and depression) and attentional control in melanoma survivors by deploying a methodology that allowed for the effects to be identified, excluding the variance related to chemotherapy. Our results broadly concurred with Calvio et al. [23], who reported that higher symptoms of anxiety and depression (assessed using an emotional symptom checklist) were associated with poorer attention (indexed by a self-reported cognitive symptom checklist) in breast cancer survivors, and Rogiers et al. [18,19], who found unique relationships between anxiety/depression and attentional problems in melanoma survivors. The results of our statistical modelling, however, showed that anxiety and depression are differentially associated with attentional control, and specifically that higher depression and trait-anxiety are related to poorer self-reported attentional control [30,31]; we found this pattern specific to individuals who reported higher cancer-related worry. As such, our results were in accord with Spada et al. [31], who found that higher worry was linked to poorer attentional problems in a non-cancer sample; yet, we noted similar results in high trait anxious melanoma survivors who reported high depression. Meanwhile, those who reported both high levels of trait anxiety and worry recounted poorer attentional control in the presence of low depression.

Our results differ from some previous studies in cancer survivors, such as those of Krolak et al. [21] and Areklett [22], who found no relationship between anxiety, depression, and attention. This discrepancy might have resulted from methodological differences. Specifically, the current study used a discrete and valid measure of attentional control (i.e., ACS), whereas Krolak et al. and Areklett et al. used measures of cognitive problems more broadly (i.e., FACT-Cog and CSC, respectively). Similarly, we used measures of trait anxiety and situational or cancer-related worry, whereas Krolak et al. and Areklett et al. assessed clinical symptoms of anxiety. Moreover, these earlier works were not able to separate the influences of anxiety and depression, which was addressed by our statistical modelling. Hence, their findings might have been influenced by the confounding effect of depression rather than anxiety itself, or the confounding effect of anxiety rather than depression itself.

Our results offer support for the central assumptions of attentional control theory [24] and are consistent with the hypothetical assumption that anxiety consumes cognitive resources and impairs the ability to maintain and control attention. Importantly, the present results demonstrate the importance of including the unique contribution of comorbid depression when examining the anxiety–attention link, such that anxiety can boost; yet, depression can attenuate attentional control, at least in melanoma cancer survivors. As such, future iterations of attentional control theory may consider adding depression to the model.

## 5. Strength, Limitations, and Future Research

Our work addressed an important gap in the existing literature and adds much-needed knowledge about melanoma survivorship. We deployed a sample of melanoma survivors, and used a theoretical framework and empirical studies to guide our hypotheses. Our study, however, is not without some limitations. We used a subjective (albeit, a well-used and validated measure) of attentional control. Future studies using melanoma survivors are needed to corroborate our findings using behavioral measures of attentional control [17], which would enable an assessment of both the quality (effectiveness) and efficiency of task performance and remove self-report bias. Given that our study included mostly female melanoma survivors and those predominantly within their first year of diagnosis and treatment, it is possible that our results are not able to be generalized to male melanoma survivors and individuals diagnosed/treated with melanoma more than 1 year prior to the study. It is also possible that using a volunteer database meant our participants were qualitatively different (e.g., more motivated) compared to non-volunteers. While a cross-sectional design is known to be problematic, it is for future work to examine the robustness of our findings.

## 6. Implications

Our findings have implications for primary healthcare workers, such as medical practitioners (physicians), oncologists, psychiatrists and psychologists who provide holistic care to melanoma survivors. In cancer treatment, mental health often loses priority as a significant treatment aim, with medical practitioners working in their scope to rectify the biological anomaly. With a larger proportion of these individuals with melanoma surviving due to medical advancements, there is room for improvement with regard to the treatment of mental health. The clinical implications of our findings may be a catalyst for further work to explore the treatment of anxiety and depression in the melanoma trajectory. For example, our results indicate that the treatment of depression in this population would increase attentional control but only in high trait anxious individuals with high worry; meanwhile, the treatment of worry would potentially increase attentional control, but only in high trait anxious individuals who are highly depressed. It is plausible, therefore, that the somewhat new field of work known as attentional control training or cognitive control training for anxiety and depression [38,39], in conjunction with other evidenced-based treatments for emotional distress, may provide both positive affective and cognitive outcomes. We hope the present study becomes the catalyst for such work.

## 7. Conclusions

In sum, we found that depression moderates the relationship between anxiety (trait anxiety and situational cancer worry) and attentional control in melanoma survivors. The finding that depression and anxiety have different relationships with attentional control is important. Specifically, we found that, for melanoma survivors reporting lower depression, higher trait anxiety was related to higher attentional control at higher cancer worry levels but not at lower cancer worry levels. For those reporting higher depression, higher trait anxiety was associated with poorer attentional control at higher cancer worry levels but not at lower cancer worry levels. These data provide preliminary findings regarding the inter-relationships between anxiety, depression, cancer worry, and attentional control, which offer important clinical markers for healthcare professionals. Future research is required to validate the strength and reliability of these results.

## Figures and Tables

**Figure 1 healthcare-11-03097-f001:**
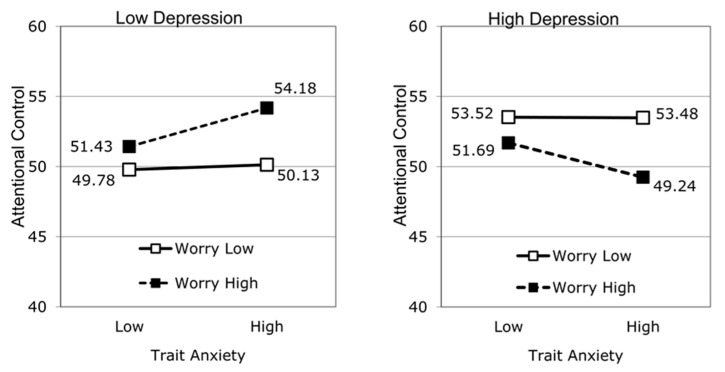
Relationship between trait anxiety, worry, depression, and attentional control. Note. Simple slopes are calculated at ±1 SD from the mean score on each of the high and low values on the predictor variables.

**Table 1 healthcare-11-03097-t001:** Demographics of the Sample of Melanoma Survivors (*N* = 187).

Characteristic		*M*_age_ (*SD*_age_)	*N* (Percent)
Age	Female age range 18–57 years Male age range 31–58 years	36.54 years (4.94) 48.62 years (12.12)	174 (93%) 13 (7%)
Depression	Normal		15 (8%)
	Mild		1 (0.5%)
	Moderate		3 (1.6%)
	Severe		2 (1.1%)
	Extremely Severe		166 (88.8%)
Marital status	Single		9 (4.8%)
	Defacto		1 (0.5%)
	Married		177 (94.7%)
	Other		0 (0.0%)
Time since diagnosis	1 year		167 (89.3%)
	2 years		8 (4.3%)
	3 years		3 (1.6%)
	4 years		4 (2.1%)
	5 years		3 (1.6%)
	6 years		2 (1.1%)
Treatment	Chemotherapy		165 (88.2%)
	Radiation		166 (88.8%)
	Surgery		187 (100%)
	Drug therapy		168 (89.8%)
Time since treatment	1 year		161 (86.1%)
	2 years		12 (6.4%)
	3 years		5 (2.7%)
	4 years		4 (2.1%)
	5 years		2 (1.1%)
	6 years		1 (0.5%)
	7 years		2 (1.1%)

**Table 2 healthcare-11-03097-t002:** Means, standard deviations, zero-order and inter-correlations of trait anxiety, worry, depression, and attentional control.

	Cronbach’s Alpha	*M*	*SD*	Trait Anxiety	Worry	Depression
Trait Anxiety	0.94	33.35	6.95			
Worry	0.86	64.17	5.16	0.73 ***		
Depression	0.92	33.92	8.38	0.71 ***	0.75 ***	
Attentional Control	0.75	51.23	2.83	0.21 **	0.20 **	0.27 ***

*** *p* < 0.001, ** *p* < 0.01.

**Table 3 healthcare-11-03097-t003:** Unstandardized coefficients, *t* values, *p* values, and 95% confidence intervals for all variables at each step for attentional control.

		Unstandardised Coefficients		*t*	Sig.	95% Confidence Intervals for B	
		B	Std. Error			Lower Bound	Upper Bound
Step 1	(Constant)	50.28	0.36	139.00	0.000	49.57	51.00
	Chemotherapy	1.16	0.36	3.20	0.002	0.444	1.87
Step 2	(Constant)	50.91	0.21	238.33	0.000	49.81	50.64
	Chemotherapy	0.40	0.55	0.73	0.469	−0.68	1.47
	Trait Anxiety	0.01	0.05	0.24	0.81	−0.08	0.10
	Worry	−0.02	0.07	−0.27	0.784	−0.15	0.11
	Depression	0.07	0.04	1.64	0.102	−0.02	0.16
Step 3	(Constant)	51.39	0.56	91.93	0.000	50.29	52.50
	Chemotherapy	0.14	0.56	0.24	0.809	−0.963	1.23
	Trait Anxiety	0.01	0.05	0.30	0.765	−0.08	0.11
	Worry	−0.04	0.10	−0.38	0.701	−0.22	0.15
	Depression	−0.01	0.05	−0.16	0.876	−0.12	0.10
	Trait Anxiety × Worry	0.01	0.01	0.60	0.547	−0.02	0.03
	Trait Anxiety × Depression	−0.01	0.01	−2.37	0.019	−0.02	−0.00
	Worry × Depression	0.00	0.01	0.11	0.913	−0.01	0.02
Step 4	(Constant)	52.11	0.59	88.73	0.000	50.95	53.27
	Chemotherapy	−0.52	0.58	−0.90	0.368	−1.66	0.62
	Trait Anxiety	0.01	0.05	0.24	0.809	−0.08	0.10
	Worry	−0.01	0.09	−0.10	0.922	−0.19	0.17
	Depression	0.04	0.05	0.66	0.512	−0.07	0.14
	Trait Anxiety × Worry	0.00	0.01	−0.02	0.982	−0.02	0.02
	Trait Anxiety × Depression	−0.01	0.01	−2.43	0.016	−0.02	−0.00
	Worry × Depression	−0.03	0.01	−2.61	0.101	−0.06	−0.01
	Trait Anxiety × Worry × Depression	0.00	0.00	−3.24	0.001	−0.00	−0.00

## Data Availability

The data presented in this study are available upon request from the corresponding author.

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
