# Peer review of "Depression Moderates the Relationship between Trait Anxiety, Worry and Attentional Control in Melanoma Survivors"

_healthcare, 2023, doi:10.3390/healthcare11233097_

Round 1
Reviewer 1 Report
Comments and Suggestions for Authors
1. Please add more keywords (up to 10) and sort them alphabetically.
2. The Introduction section is too long compared to other sections, especially the Discussion. In general, the paper is imbalanced. Please shorten your discussion. In some cases, you presented too much unimportant details. Please add a psychological context here, and clearly indicate the relevance of this study.
3. Table 1, last line, add "years".
4. Please indicate alpha Cronbach's for your questionnaires your study in your Table 2, as these internal consistency reliability coefficients are a part of results.
5. Lines 155 and below: (2, 4, 5, 6, 7, 9, 12, 13, 14, 15 - redundancy. There is no need to present this.
6. Typos, e.g. line 165, etc. Use ; between statements when you present two different examples of statements in the measure descriptions.
7. When the study was conducted? Pre-registration?
8. Please indicate the percentage of positively screened respondents in the depression scale.
9. In general, the study is conducted correctly. However, the novelty of this study is somewhat poor and in general the paper is not very interesting. Try to provide more solid hypotheses related to the needs of medicine and society, without too much focusing on extra statistical analyses. It seems that the paper is appeared to be more statistical than psychological. Statistics is just only a tool in psychology, and no more.
Author Response
Thank you or your valuable feedback.

Reviewer 2 Report
Comments and Suggestions for Authors
This is an interesting study with some points that need to be address to improve the quality of the manuscript. I report my comments/suggestions:
- the aims of the study should be better explained, the authors reported only a general aim that should be deepen to justify the analysis and the hypothesis
-the authors should include more ditails on participants selection, the potential bias of use a volonteer database should be reported. Moreover I suggest to include which information have been given to participants
-about the sample size did the authors performed a power analysis to evaluate the participants needed? If not at least include a post hoc power analysis for the regression models tested
-I have understand the authors' choice to include chemiotherapy as covariant but I don't know why the authors did not include other relevant variables as age and gender since I think that are strongly associated to the dimensions investigated. Please consider to include other variables as covariants
- I suggest to enrich the discussion
-Limits: include self report measures, other confounders, sample size (I do not think it is in absolut a large sample), cross sectional study
Comments on the Quality of English LanguageCheck for typos
Author Response
Thank you for your valuable feedback.

Round 2
Reviewer 1 Report
Comments and Suggestions for Authors
The paper was improved satisfactorily.
Use "Cronbach's alpha" instead of incorrect "Cronbach alpha".
Add this sentence to the data analysis section as this is now in inadequate place within the participants description: For our linear regression with eight predic- 147 tors, an a priori power analysis using G*Power indicated that a sample of 160 would 148 achieve a power of 0.95 at p<.05, considering a small effect size (d = .15).
The Procedure should be placed before the Participants as conducting research, you plan a procedure before recruiting participants. Thus, be logical when presenting aspects of your study.
As for internal consistency reliability, favorable is not a good word here. See Groth-Marnat (2009) with acceptable, good and excellent classification.
Author Response
Thank you for your feedback.

Reviewer 2 Report
Comments and Suggestions for Authors
I found the revised version of the manuscript improved and clearer for readers. I have only a last minor point that do not required further revision.
The power analysis for regression with 8 predictors required 160 subject as total sample size with an effect f2 .15 that is medium and not small (the authors reported d. 15 small)
Author Response
Thank you for your feedback.
